# Assessing the Impact of Health Education Intervention on Asthma Knowledge, Attitudes, and Practices: A Cross-Sectional Study in Erbil, Iraq

**DOI:** 10.3390/healthcare11131886

**Published:** 2023-06-29

**Authors:** Karwan Bahram Maulood, Mohammad Khan, Syed Azhar Syed Sulaiman, Amer Hayat Khan

**Affiliations:** 1School of Pharmaceutical Sciences, Universiti Sains Malaysia, Gelugor 11700, Malaysia; 2School of Medical Sciences, Universiti Sains Malaysia, Kota Bharu 16150, Malaysia; drmohammadkhan1001@gmail.com; 3New Age Health Science Research Center, Aturar Dipu, Muradpur, Chittagong 4231, Bangladesh

**Keywords:** asthma, Erbil, health education, knowledge, attitudes, practice

## Abstract

Asthma causes chronic coughing, wheezing, dyspnea, and chest pressure. This study assessed asthmatic patients’ knowledge, attitudes, and practice (KAP) of bronchial asthma and proper education on its meaning, risk factors, symptoms, diagnosis, management, and prevention practices. We performed a cross-sectional interventional asthma KAP survey in Erbil, Iraq. We adapted a validated study questionnaire from KAP studies in other nations to the Erbil situation and culture. In Erbil, Kurdistan, Iraq, two major hospitals’ asthma clinics were studied. We chose 250 asthmatic patients from October 2018 to July 2019. Health education was comprehensive. The health education program used a Kurdish PowerPoint with a printout. Twenty-five groups received two weeks of one-hour health education pre-intervention. Each group was questioned before, 2 weeks after, and 12 weeks after health education. All data were analyzed by SPSS v26. The mean age of the respondents was 37.52 ± 15.16, with 48.7% of the respondents having a positive family history of asthma. After 2 weeks of health education intervention, respondents had a higher knowledge score and positive attitudes compared to pre-education, and after 12 weeks of education, there was a significant difference (*p* < 0.001) with improvised prevention practice. Health education programs led to considerable improvements in asthmatic patients’ knowledge, attitudes, and practices regarding their condition. After receiving health education for a period of two weeks, the majority of the participants answered correctly regarding asthma, its causes, and the elements that trigger asthma attacks.

## 1. Introduction

Asthma is a chronic inflammatory condition of the airways, including mast cells, eosinophils, neutrophils (particularly in abrupt onset, deadly exacerbations, occupational asthma, and smokers), T lymphocytes, macrophages, and epithelial cells [1]. In sensitive individuals, this inflammation causes repeated coughing, wheezing, dyspnea, and chest tightness. Widespread but mild airflow blockages can be reversed spontaneously or medically. Increases in asthma prevalence, morbidity, and mortality have intensified public health concerns [2]. Clinical manifestations of asthma can be controlled with appropriate treatment. There should be only occasional symptoms and no serious asthma attacks. Patients with asthma who are hospitalized have a lower preference for decision-making autonomy [3]. Delaying the commencement of appropriate therapy during acute, severe attacks might potentially result in adverse asthma outcomes. This is a major cause for concern because patients are known to discontinue treatment unintentionally due to a lack of knowledge about the numerous treatment options [4]. When a patient’s conduct and attitudes are affected by fear, it is because they do not grasp the situation. Providing asthma patients with the education they need to manage their condition effectively is a top priority for healthcare providers. Noncompliance, insufficient severity assessment, and inadequate therapy all contribute to generally poor control [5]. Only physicians have been offering asthma education for decades to patients, which supports the necessity for additional sectors, including nongovernmental groups, the media, and health professionals, to educate patients about asthma [6]. Denial about having a chronic condition, poor knowledge of the disease process and medication use, as well as poor comprehension of how to utilize inhalers, are all variables that contribute to asthma morbidity [5,6].

An interactive learning experience that changes patients’ knowledge and health behaviors via teaching, counseling, and behavior modification tactics enables patients to actively participate in their health care. Most experts believe asthma education improves patient understanding, but other health effects vary [7]. Lack of these skills increases asthma-related hospitalizations and ER visits. Morbidity, education, knowledge, and sickness behaviors remain mysteries. Patients were taught about a disease’s origin, treatments, and triggers [8]. They will all tell you that they are overwhelmed by the amount of information out there on asthma. Generally, asthma education programs teach participants how to manage their asthma with a doctor (self-management) or both [9]. A comprehensive asthma self-management program is required. One of the simplest and most cost-effective ways to educate people about asthma is to simply provide them with information on the disease and how to treat it. Either a hospital or a community setting can easily undertake this study [10]. An interactive or non-interactive method of delivering asthma information can be used. Lectures, audiovisual presentations, and group discussions are all examples of forms of interactive learning that may be used in either individual or group sessions with a teacher. Other examples include role-playing, project- or assignment-based learning, participatory learning, and the case method for developing problem-solving skills [11,12].

The present study, based on the Knowledge, Attitude, and Practice Perception (KAP) framework [6], aims to investigate patients’ understanding of risk factors associated with asthma development and exacerbation, as well as strategies to support individuals affected by this condition. These educational sessions are designed to enhance patients’ knowledge about asthma, mitigate risk factors, and minimize medication-related side effects [4]. Notably, a previous study demonstrated a significant positive correlation between asthma knowledge, social support, and variability in self-management behaviors [5]. Consequently, the objective of our research is to evaluate the level of knowledge among individuals with asthma concerning bronchial asthma and to provide appropriate education to enhance their understanding of its definition, risk factors, signs and symptoms, diagnostic approaches, management strategies, and preventive measures.

## 2. Materials and Methods

### 2.1. Design and Ethics

A cross-sectional interventional survey was conducted in Erbil, Iraq, utilizing a questionnaire to assess the knowledge, attitude, and practice (KAP) of patients diagnosed with asthma. The questionnaire employed in this study was validated and based on established methodologies used in KAP studies conducted in various countries. To ensure cultural and contextual relevance, the questionnaire was adapted to align with the specific circumstances in Erbil. Ethical approval for this study was obtained from the Ethics Committee for Research Involving Human Subjects at Universiti Sains Malaysia (USM), the Ministry of Health’s Ethics Committee for Research Involving Human Subjects in Iraq, as well as from the authorities at Rizgary Teaching Hospital and Hawler Medical University, where the study was conducted.

### 2.2. Participants

The study was conducted in the asthma clinics of two tertiary hospitals in Erbil, Kurdistan, Iraq. We selected 250 patients with asthma who presented at a hospital between 1 October and 31 July and had been diagnosed with asthma for at least 3 months. The diagnoses must have conformed with the guidelines for the diagnosis and management of asthma issued by the Ministry of Health, Iraq.

### 2.3. Questionnaire

Vallerand’s technique [13] validates Gare’s KAPQ across cultures [14]. Seven steps: “1. Preparation of a preliminary version; 2. Evaluation and revision; 3. Pretest (verification of item clarity by the target population); 4. Concurrent and content validity; 5. Reliability; 6. Construct validity; 7. Concurrent validity, construct validity, reliability, and responsiveness to questionnaire design changes”.

### 2.4. Preparation

The first draft of the Kurdish KAPQ was developed after the English version was translated by a professional translator with experience in the medical field. Kurdish-speaking asthma specialists reviewed each item of the KAPQ to ensure that it contained the correct medical terminology. As a final step, a professional translator performed a back-translation to see how the two translations differed in the original language.

### 2.5. Evaluation and Modification of Content

Two other bilingual asthma specialists and a psychometric methodology expert evaluated the preliminary KAPQ, which was conducted by a committee of four researchers. Content validity was determined after two levels of evaluation. The committee first compared the original and back-translated English versions of each item to ensure that the meaning was consistent for each pair of items. Secondly, they checked the final Kurdish version (KAPQ) to make sure that there were no jargon-laden technical terms in it.

### 2.6. Pretest of the PAKQ

Kurdish-speaking asthmatics (n = 25) were invited sequentially to test the questionnaire to ensure that each item was clear, unambiguous, and written in a language they were familiar with before it was distributed to the general population. Random-probe interviews were used to accomplish this. The 42 questions were all found to be unambiguous, and it appears that patients had no difficulty deciphering their meaning. The original has not been altered in any way.

### 2.7. Validation

By conducting confirmatory factor analyses, we were able to determine the questionnaire’s factorial structure. KAPQ was found reliable with high internal consistency (T_0_ = 0.84 and T_1_ = 0.92).

### 2.8. Health Education Program

A robust health education session was prepared with the help of the Asthma and Allergy Foundation of America (AAFA) and the School of Pharmaceutical Sciences at Universiti Sains Malaysia. A PowerPoint presentation in Kurdish with a printout was used to deliver the health education program. PPT slide content and information were purchased with copyright from AAFA for patient education purposes. The presentation’s contents were organized in the following order: common definition of asthma, causes, risk factors, diagnosis, treatment modalities, use of inhalers, preventive measures, and final message.

### 2.9. Intervention

Interview development used pre- and post-test designs. Before health education, asthmatics were questioned on their preventative knowledge, attitudes, and practices. Health education was provided for two weeks. Twenty-five patients were grouped together. Two sets of kids had a one-hour education session each morning and afternoon. Each ten-patient group had a PI-carrying respiratory physician assistant. The intervention was documented and communicated throughout. Every meeting ended with questions and replies for further discussion.

### 2.10. Data Collection

Initially, pre-intervention evaluation questionnaires and a cover letter describing the study objective were issued. Two hospitals conducted two-day pre-testing. A total of 142 Rizgary hospital patients and 100 Hawler hospital patients under Hawler Medical University completed pre-test questionnaires. Patients were briefed on the study and told not to discuss their questions or responses with others. They consented to fill out a questionnaire for this study. Both universities’ lecture halls hosted these events. The questionnaire could be completed in these well-lit, quiet places. The patients were encouraged to return the questionnaires quickly. The patients could easily drop off the completed questionnaires at any hall ward’s main desk. Health education followed. Thus, the patients received PPT slides and printed documents. Twenty-five groups received one-hour interventions over two weeks. Twelve weeks after health education, participants received similar questions. The patients were instructed to keep their issues private. The patients could drop off the completed questionnaires at the front desk, just like in the pre-intervention assessment.

### 2.11. Analysis of Data

Means and standard deviations were used as measures of central tendency and dispersion to describe the continuous variables. For non-normally distributed variables, the variables were described based on the median and interquartile range. Categorical measured variables were described using frequencies and percentages. A paired t-test was used for the comparison of the pre- and post-test scores.

## 3. Results

### 3.1. Patients’ Characteristics

Table 1 shows the socio-demographic characteristics of the respondents. The mean age of the respondents was 37.52 ± 15.16. More than half of the participants were male (51.6%). Nearly two-thirds were married (72.4%), while 41.6% were reported as illiterate. Housewife (28.4%) was the most common occupation, followed by worker (24.8%) and professional (20.0%), such as teacher, engineer, doctor, architect, etc. Only one-third of the respondents had smoking habits (26.9%). A total of 48.7% of respondent had a positive family history of asthma with a duration of 1–2 years (46.2%), followed by 0–12 months (39.8%).

### 3.2. Knowledge of Asthmatic Patients Regarding Asthma before Health Education

Table 2 shows how much asthma patients knew about their condition before receiving health education. All survey takers were aware that asthma is an airway disease, but most had only a superficial understanding of how it manifests. Most responders had trouble differentiating between the many factors that worsened asthma-related respiratory issues. Air pollution (named by 80% of participants) and the common cold were both named by 100% of participants. In addition, everyone present was aware that smoking is a major contributor to the onset of asthma. Participants also demonstrated a lack of familiarity with asthma care recommendations and available treatments, according to the poll results.

Table 3 shows asthmatics’ 2-week health education program results. A total of 91.2% of those who guessed correctly said asthma changes the lung and breathing tube. Wheezing (80%), time-varying (90.4%), and nighttime/morning (78.8%) asthma symptoms were correctly identified. They specified allergies (80%), air pollution (100%), colds (100%), exercise (100%), particular foods (68.4%), and incorrect notions such as “living with asthma patients causes asthma” (100%). Patients received asthma management tools. Health education greatly increased patients’ health knowledge. Table 4 demonstrates asthmatic patients’ asthma awareness after 12 weeks of health education. Most patients forget asthma information after a while. A total of 80.0% of respondents correctly recognized asthma etiology but misidentified asthma pathophysiology. Most respondents identified asthma symptoms and triggers.

Table 5 shows the comparison of the knowledge score before and after the health education intervention. After 2 weeks of health education intervention, respondents had a higher knowledge score compared to pre-education and after 12 weeks of education, with a significant difference (*p* < 0.001). There was no significant association of the knowledge score with gender, education, family history, or occupation (*p* > 0.005) (Figure 1, Figure 2, Figure 3 and Figure 4).

### 3.3. Attitudes of the Asthmatic Patients before and after Health Education Intervention

Table 6, Table 7 and Table 8 show the comparison of the attitude’s responses with the score (Table 9) before health education, 2 weeks after, and 12 weeks after health education. There is a significant difference found between the time intervals with a better positive attitude after health education (*p* < 0.005). Although there is no significant score difference observed between gender (Figure 5) and duration of asthma (Figure 6).

### 3.4. Practice of the Asthmatic Patients before and after Health Education

Table 10 shows the practice of the asthmatic patients regarding asthma before education and after 2 weeks and 12 weeks of health education intervention [8]. After 2 weeks, visiting a physician was practiced by 92%, while after 12 weeks, it was reduced to 81.9%. Nasal spray use also increased after HEI at 2 and 12 weeks. Before HEI, 99.6% of the patients were used to buying over-the-counter drugs without physician advice, which completely zeroed out after education. Although a total of 36.1% again bought over-the-counter drugs without physician advice after 12 weeks. After HEI, almost 88.0% of patients avoid house dust and smoke and also follow their doctor’s advice strictly. Physical exercise also increases after HEI, which helps them in their daily activities. A total of 87.6% of patients did not forget to take medication in the last two weeks after 2 weeks of HEI, and 37.2% reported that they forgot after 12 weeks. A total of 74.8–80.8% of patients reported that they were changing medications if their asthma worsened after consulting the proper channel. After HEI, patients were more concerned about avoiding smoke and dust by removing them with a fan or more cautiously using deodorant to avoid sudden asthma attacks.

## 4. Discussion

The Global Initiative for Asthma [9] and the asthma guidelines for prevention and treatment written by an expert panel from the National Heart, Lung, and Blood Institute emphasize the importance of promoting a standardized classification of asthma treatment [15]. In this analysis of the KAP of Erbil patients with asthma, a wide gap was observed between recommended and actual practices, and their overall asthma-related knowledge was insufficient at baseline. Furthermore, asthma-related knowledge was associated with deficiencies in the care process. After successful implementation of the health education program, overall KAP improved. The importance of asthmatic habits in controlling disease cannot be overstated [16]. This research was carried out in Erbil with the purpose of determining the levels of knowledge, attitudes, and practices held by adult patients who suffer from asthma. The questionnaire was based on knowledge of the fundamental pathophysiology of the disease, symptoms, triggering factors, precipitating causes, medication, and management of asthma, all of which are crucial for individuals who have chronic asthma. In addition to that, it evaluated the behaviors that patients need to follow in order to reduce the likelihood of future asthma exacerbations.

Our study found that all participants knew that asthma is an airway illness. However, most did not know asthma’s causes, symptoms, or triggers. Asthma guidelines and treatment alternatives were unfamiliar to survey respondents. This study found that prior asthma education does not improve asthma control or quality of life. This matches prior studies [17]. Comparing asthma information gained from personal experience to that gained via active education, Meyer et al. observed that personal asthma knowledge differed more than active education [18]. However, knowledge still reduces asthma morbidity and mortality. Other studies also found a correlation between asthma knowledge and asthma severity [19].

Asthma attitudes and beliefs that promote excellent health and medication compliance for effective disease management were also considered [20]. This ensured optimum illness management. Most asthmatics know little about the disease and its treatments. Higher patient education may have affected their knowledge. After health education, the time intervals with a more positive attitude differed statistically (*p* < 0.005). Due to the majority of participants not having completed secondary education, it is suggested. However, college-educated people understood their ailment and treatment better. Our findings reflect prior data suggesting better education is strongly connected with asthma knowledge [21].

Asthma control improves the quality of life. Effective asthma management may benefit from non-educational asthma control methods. Health education raised patients’ knowledge significantly. After two weeks of health education intervention, respondents had a significantly higher knowledge score than before and after twelve weeks (*p* < 0.001). After two weeks of health education, 91.2% could identify asthma pathogenesis. One hundred percent of participants also knew the causes that worsen asthmatic episodes. Health care providers should empower asthmatics to manage their own health and have confidence in doing so. Patients should help establish a self-management approach [22]. Health education teaches people how to handle symptoms and well-being. Patients’ asthma and treatment knowledge will rise. If health practitioners consider patients’ beliefs and goals to meet their needs, such programs will gain credibility. College-educated patients had this compared to non-graduated patients [23]. A recent comprehensive study found that patients still trust doctors more than social media and the internet for health information [24].

Many patients had misperceptions about asthma and related illnesses before receiving health education. This is another impediment to proper information, along with illiteracy and ignorance. Sodhi and colleagues researched how bronchial asthma patients share knowledge, attitudes, and behaviors. (In terms of asthma beliefs, 64% of patients were uninformed of the etiology of their disease; 16.3% thought it was caused by allergens; 8.7% thought it was genetics; and 3.6% thought it was a curse from God.) Kumar et al. found that several patient characteristics affected asthma knowledge [25]. Despite prior studies, there was a significant age-related difference in patient knowledge. We postulate that younger age groups have less life experience with asthma, while elderly patients (those older than 60 years) have less asthma knowledge due to forgetfulness and memory impairment [26]. Inhaled medications are misunderstood. Most asthmatics use inhalers, but some say pills work better. This misperception may make asthma medication compliance harder. It is also associated with the shift away from inhaler medicines to other treatments [27].

The international asthma guidelines recommend collaboration between asthmatics and their health care providers [28]. This method should provide patients with the knowledge, confidence, and abilities to manage their asthma. Self-management has been shown to lower asthma morbidity. However, misperceptions regarding asthma and inhaled medicine may lead to poor self-control [29]. These guidelines also prescribe regular preventive inhaled medicine and bronchodilator therapy to prevent and treat chronic asthma [30]. In impoverished countries, inhaled corticosteroids reduce hospital admissions and ER visits by up to 80% [31,32]. Some research from other territories supports our findings on adult asthma practice and this session’s variables. Higher-educated patients can read asthma instructions and execute better self-care. Higher-income patients have easier access to health services. Experienced asthmatics can also better manage their symptoms. It also reduces asthma exacerbations, which is consistent with an earlier study [33]. Patients who receive asthma education from their healthcare providers and community pharmacists have better asthma management [34]. Patients who can self-manage and have a high quality of life have fewer asthma attacks and can manage their disease more efficiently and easily than those who cannot [35].

Despite recognizing symptoms and trigger points, some individuals lose their asthma expertise after twelve weeks. A recent study suggests that patients should continue health education for long-term asthma control [36]. According to the findings of a recent study, it is important for patients to continue their health education for long-term asthma control. Education boosts patients’ and providers’ confidence and competence, and it improves practice [37]. Our study clearly links patient education with results. Asthma management may benefit from brochures, books, and self-practice cards [34,35].

## 5. Conclusions

Health education programs led to considerable improvements in asthmatic patients’ knowledge, attitudes, and practices regarding their condition. After receiving health education for a period of two weeks, the majority of the participants answered correctly regarding asthma, its causes, and the elements that trigger asthma attacks. There is a statistically significant difference (*p* < 0.001) between the respondents’ knowledge scores before and after receiving health education for a period of 12 weeks, but after receiving health education for only two weeks, respondents’ knowledge scores are significantly higher, and their positive attitudes are significantly improved.

## Figures and Tables

**Figure 1 healthcare-11-01886-f001:**
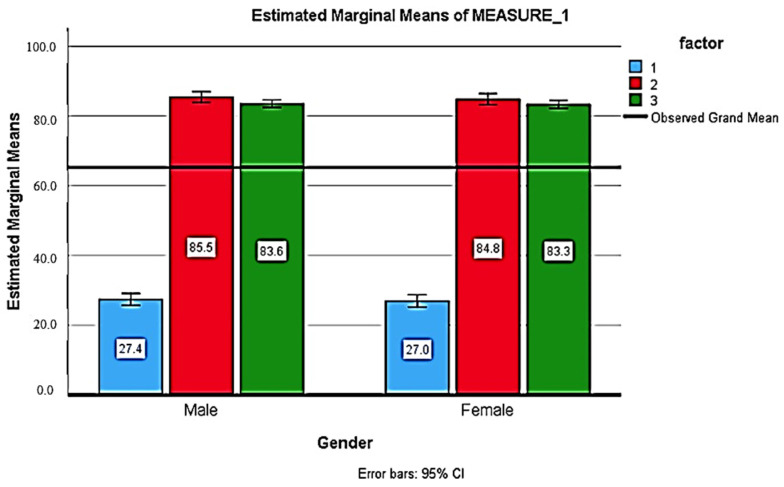
Comparison of the knowledge score before, after 2 weeks and 12 weeks of health education between the genders (Blue: baseline; Red: after 2 weeks; Green: after 12 weeks).

**Figure 2 healthcare-11-01886-f002:**
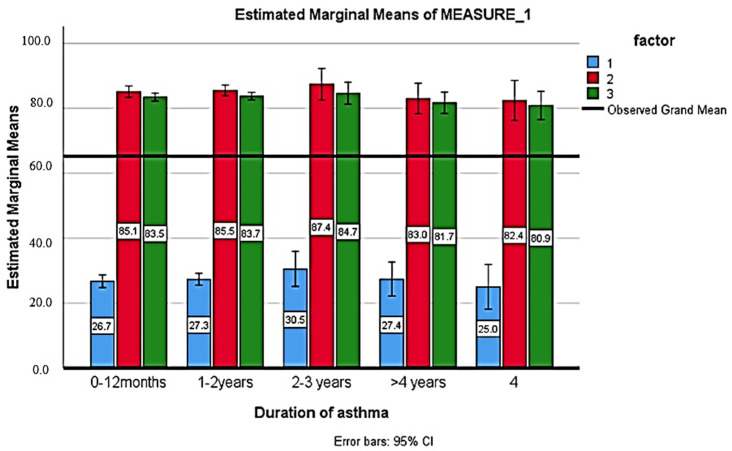
Comparison of the knowledge score before, after 2 weeks and 12 weeks of health education between the duration of asthma (Blue: baseline; Red: after 2 weeks; Green: after 12 weeks).

**Figure 3 healthcare-11-01886-f003:**
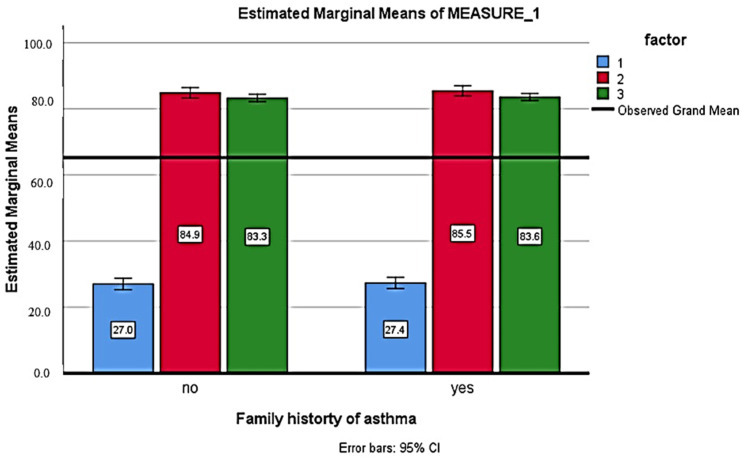
Comparison of the knowledge score before, after 2 weeks and 12 weeks of health education with family history (Blue: baseline; Red: after 2 weeks; Green: after 12 weeks).

**Figure 4 healthcare-11-01886-f004:**
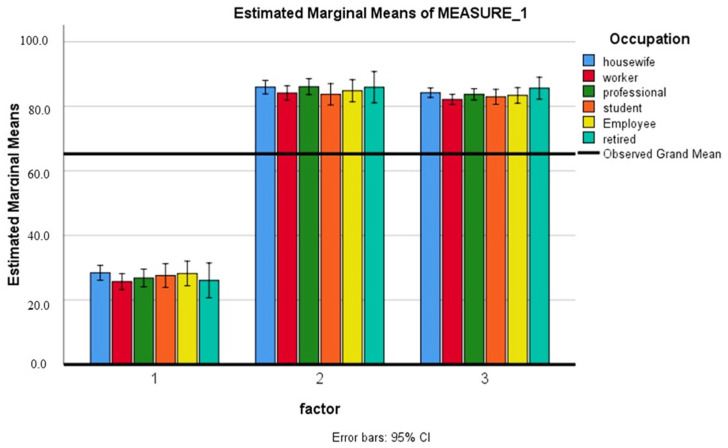
Comparison of the knowledge score before, after 2 weeks and 12 weeks of health education with occupation (1: baseline; 2: after 2 weeks; 3: after 12 weeks).

**Figure 5 healthcare-11-01886-f005:**
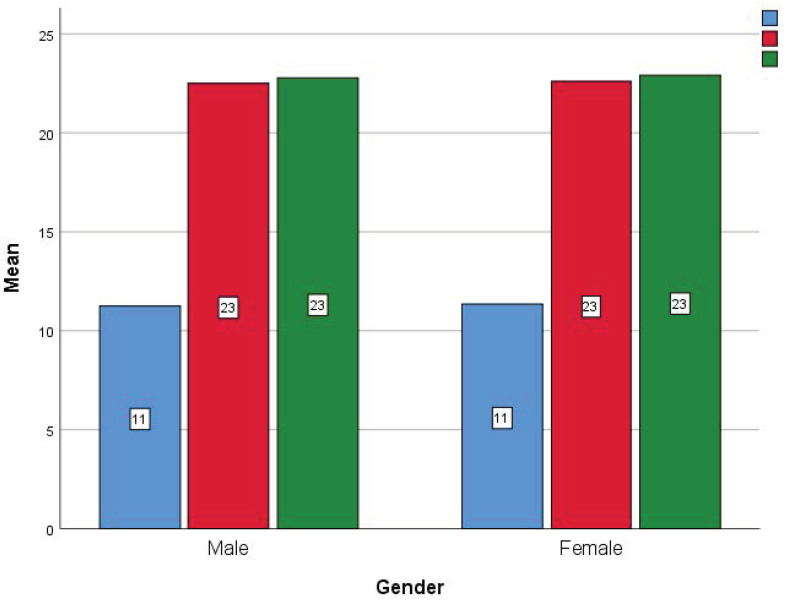
Comparison of the attitudes score before, after 2 weeks and 12 weeks of health education between the genders (Blue: baseline; Red: after 2 weeks; Green: after 12 weeks).

**Figure 6 healthcare-11-01886-f006:**
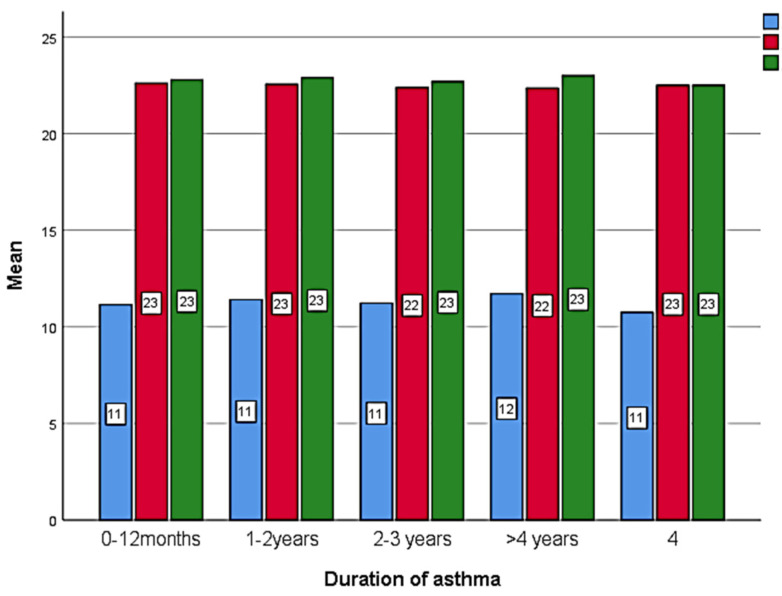
Comparison of the attitudes score before, after 2 weeks and 12 weeks of health education between the duration of asthma (Blue: baseline; Red: after 2 weeks; Green: after 12 weeks).

**Table 1 healthcare-11-01886-t001:** Characteristics of the respondents.

Variable	Frequency, N (%)
**Age ***	37.52 ± 15.16
**Gender**	
Male	129 (51.6)
Female	121 (48.4)
**Marital status**	
Married	181 (72.4)
Single	69 (27.6)
**Educational qualification**	
Illiterate	104(41.6)
Primary/secondary	83 (33.2)
College/University	63 (25.2)
**Occupation**	
Housewife	71 (28.4)
Worker	62 (24.8)
Professional	50 (20.0)
Student	28 (11.2)
Employee	26 (10.4)
Retried	13 (5.2)
**Smoking habit**	
Yes	67 (26.9)
No	182 (73.1)
**Duration of asthma**	
0–12 months	99 (39.8)
1–2 years	115 (46.2)
2–3 years	13 (5.2)
>4 years	22 (8.8)
**Family history**	
Yes	129 (48.7)
No	121 (45.7)

* Mean.

**Table 2 healthcare-11-01886-t002:** Knowledge of asthmatic patients regarding asthma before health education.

Item Variable	Agree	Disagree	No Opinion
Asthma is a chronic inflammatory disorder of airways.	100 (100)	0 (0.0)	0 (0.0)
In asthma, breathing tubes in lungs become narrow due to mucus (sputum) collection.	0 (0.0)	101 (40.4)	149 (59.6)
In asthma, breathing tubes in lungs become narrow due to tightening of muscles around them.	176 (70.4)	74 (29.6)	0 (0.0)
In asthma, breathing tubes in lungs become narrow due to swelling of their walls.	49 (19.6)	82 (32.8)	119 (47.6)
Symptoms of asthma are breathing difficulty with wheezing sound.	0 (0.0)	127 (50.8)	123 (49.2)
Asthma symptoms vary from time to time, less at some times and more at other times.	45 (18.0)	151 (60.4)	54 (21.6)
Asthma symptoms are more likely to occur at night or early morning.	73 (29.2)	52 (20.8)	125 (50.0)
Asthma symptoms can be caused by:			
(a) Allergy	76 (30.4)	174 (69.6)	0 (0.0)
(b) Air pollution (dust)	200 (80.0)	26 (10.4)	24 (9.6)
(c) Living with asthma patient	200 (80.0)	0 (0.0)	50 (20.0)
(d) Common cold	250 (100.0)	0 (0.0)	0 (0.0)
(e) Exercise	25 (10.0)	101 (40.4)	124 (49.6)
(f) Certain food	0 (0.0)	132 (52.8)	118 (47.2)
(g) Without obvious reason	105 (42.0)	0 (0.0)	145 (58.0)
Smoking makes asthma worse.	250 (100.0)	0 (0.0)	0 (0.0)
Asthma medicine can be taken as tablet/syrup/inhalers.	7 (2.8)	118 (47.2)	125 (50.0)
The best way to take asthma medicine is inhalation.	32 (12.8)	50 (20.0)	168 (67.2)
Asthma medicines are usually of two types: one to give immediate relief and the other to prevent symptoms.	7 (2.8)	0 (0.0)	243 (97.2)
Most effective drugs for controlling asthma are called steroids.	102 (40.8)	0 (0.0)	148 (59.2)
Inhalers are free from significant side effects.	169 (67.6)	51 (20.40	30 (2.0)
Asthma medicine has to be taken until symptoms persist then can be stopped.	0 (0.0)	107 (42.8)	143 (57.2)
Asthma medicine has to be taken even after symptoms are no longer there, until your doctor advises you to stop.	50 (20.0)	50 (20.0)	150 (60.0)

**Table 3 healthcare-11-01886-t003:** Knowledge of asthmatic patients regarding asthma after 2 weeks of health education.

Item Variable	Agree	Disagree	No Opinion
Asthma is a chronic inflammatory disorder of airways.	250 (100.0)	0 (0.0)	0 (0.0)
In asthma, breathing tubes in lungs become narrow due to mucus (sputum) collection.	0 (0.0)	228 (91.2)	22 (8.8)
In asthma, breathing tubes in lungs become narrow due to tightening of muscles around them.	0 (0.0)	198 (79.2)	52 (20.8)
In asthma, breathing tubes in lungs become narrow due to swelling of their walls.	228 (91.2)	0 (0.0)	22 (8.8)
Symptoms of asthma are breathing difficulty with wheezing sound.	200 (80.0)	50 (20.0)	0 (0.0)
Asthma symptoms vary from time to time, less at some times and more at other times.	226 (90.4)	24 (9.6)	0 (0.0)
Asthma symptoms are more likely to occur at night or early morning.	197 (78.8)	53 (21.2)	0 (0.0)
Asthma symptoms can be caused by:			
(a) Allergy	200 (80.0)	50 (20.0)	0 (0.0)
(b) Air pollution(dust)	250 (100)	0 (0.0)	0 (0.0)
(c) Living with asthma patient	0 (0.0)	250 (100.0)	0 (0.0)
(d) Common cold	250 (100.0)	0 (0.0)	0 (0.0)
(e) Exercise	250 (100.0)	0 (0.0)	0 (0.0)
(f) Certain food	171 (68.4)	79 (31.6)	0 (0.0)
(g) Without obvious reason	0 (0.0)	250 (100.0)	0 (0.0)
Smoking makes asthma worse.	250 (100.0)	0 (0.0)	0 (0.0)
Asthma medicine can be taken as tablet/syrup/inhalers.	250 (100.0)	0 (0.0)	0 (0.0)
The best way to take asthma medicine is inhalation.	248 (99.2)	2 (0.8)	0 (0.0)
Asthma medicines are usually of two types: one to give immediate relief and the other to prevent symptoms.	249 (99.6)	0 (0.0)	1 (0.4)
Most effective drugs for controlling asthma are called steroids.	177 (70.8)	73 (29.2)	0 (0.0)
Inhalers are free from significant side effects.	0 (0.0)	199 (79.6)	51 (20.4)
Asthma medicine has to be taken until symptom persist then can be stopped.	0 (0.0)	130 (52.0)	120 (48.0)
Asthma medicine has to be taken even after symptoms are no longer there, until your doctor advises you to stop.	224 (89.6)	26 (10.4)	0 (0.0)

**Table 4 healthcare-11-01886-t004:** Knowledge of asthmatic patients regarding asthma after 12 weeks of health education.

Item Variable	Agree	Disagree	No Opinion
Asthma is a chronic inflammatory disorder of airways.	250 (100.0)	0 (0.0)	0 (0.0)
In asthma, breathing tubes in lungs become narrow due to mucus (sputum) collection.	203 (81.2)	47 (18.8)	0 (0.0)
In asthma, breathing tubes in lungs become narrow due to tightening of muscles around them.	101 (40.4)	149 (59.6)	0 (0.0)
In asthma, breathing tubes in lungs become narrow due to swelling of their walls.	218 (87.2)	32 (12.8)	0 (0.0)
Symptoms of asthma are breathing difficulty with wheezing sound.	200 (80.0)	24 (9.6)	26 (10.4)
Asthma symptoms vary from time to time, less at some times and more at other times.	156 (62.4)	46 (18.4)	48 (19.2)
Asthma symptoms are more likely to occur at night or early morning.	175 (70.0)	0 (0.0)	75 (30.0)
Asthma symptoms can be caused by:			
(a) Allergy	202 (80.4)	25 (10.0)	23 (9.2)
(b) Air pollution(dust)	250 (100.0)	0 (0.0)	0 (0.0)
(c) Living with asthma patient	25 (10.0)	200 (80.0)	25 (10.0)
(d) Common cold	250 (100)	0 (0.0)	0 (0.0)
(e) Exercise	175 (70.0)	49 (19.6)	26 (10.4)
(f) Certain food	199 (79.6)	0 (0.0)	51 (20.4)
(g) Without obvious reason	106 (42.4)	119 (47.6)	25 (10.0)
Smoking makes asthma worse.	243 (97.2)	5 (2.0)	2 (0.8)
Asthma medicine can be taken as tablet/syrup/inhalers.	245 (98.0)	1 (0.4)	4 (1.6)
The best way to take asthma medicine is inhalation.	200 (80.0)	43 (17.2)	7 (2.8)
Asthma medicines are usually of two types: one to give immediate relief and the other to prevent symptoms.	184 (73.6)	0 (0.0)	66 (26.4)
Most effective drugs for controlling asthma are called steroids.	207 (82.4)	43 (17.2)	0 (0.0)
Inhalers are free from significant side effects.	0 (0.0)	177 (70.8)	73 (29.2)
Asthma medicine has to be taken until symptoms persist then can be stopped.	142 (56.8)	108 (43.2)	0 (0.0)
Asthma medicine has to be taken even after symptoms are no longer there, until your doctor advises you to stop.	175 (70.0)	75 (30.0)	0 (0.0)

**Table 5 healthcare-11-01886-t005:** Comparison of the knowledge score before, after 2 weeks and 12 weeks of the health education.

Variable	Mean	95% CI		Χ^2^[df]	F[df]	*p* Value
		Lower	Upper			
**Pre score**	27.19 ± 9.83	25.96	28.41	175 (1, 63)	9283.6 (1, 249)	**<0.001**
**After 2 weeks**	85.18 ± 8.8	84.08	86.28			
**After 12 weeks**	83.4 ± 6.25	82.68	84.24			

Chi-square test and RMANOVA test were carried out.

**Table 6 healthcare-11-01886-t006:** Attitudes of the asthmatic patients before health education intervention.

Variable	Strongly Agree	Agree	Neutral	Disagree	Strongly Disagree
If one person has asthma, then all of the families are likely to have asthma as well.	43 (17.2)	139 (55.6)	38 (27.2)	0 (0.0)	0 (0.0)
Asthma is contagious.	139 (55.6)	141 (44.4)	0 (0.0)	0 (0.0)	0 (0.0)
People with asthma cannot do as much physical exercise as other people.	0 (0.0)	50 (20.0)	200 (80.0)	0 (0.0)	0 (0.0)
Asthma can be cured.	68 (27.2)	68 (27.2)	71 (28.4)	43 (17.2)	0 (0.0)
Asthma cannot be controlled.	43 (17.2)	136 (54.4)	71 (28.4)	0 (0.0)	0 (0.0)

**Table 7 healthcare-11-01886-t007:** Attitudes of the asthmatic patients after 2 weeks of health education intervention.

Variable	Strongly Agree	Agree	Neutral	Disagree	Strongly Disagree
If one person has asthma, then all of the families are likely to have asthma as well.	0 (0.0)	0 (0.0)	0 (0.0)	182 (72.8)	68 (27.2)
Asthma is contagious.	0 (0.0)	0 (0.0)	0 (0.0))	182 (72.8)	68 (27.2)
People with asthma cannot do as much physical exercise as other people.	0 (0.0)	0 (0.0)	0 (0.0))	182 (72.8)	68 (27.2)
Asthma can be cured.	71 (28.4)	179 (71.6)	0 (0.0))	0 (0.0)	0 (0.0)
Asthma cannot be controlled.	0 (0.0))	0 (0.0))	0 (0.0))	43 (17.2)	207 (82.8)

**Table 8 healthcare-11-01886-t008:** Attitudes of the asthmatic patients after 12 weeks of health education intervention.

Variable	Strongly Agree	Agree	Neutral	Disagree	Strongly Disagree
If one person has asthma, then all of the families are likely to have asthma as well.	0(0.0))	0(0.0))	0(0.0))	68(27.2)	182(72.8)
Asthma is contagious.	0(0.0))	0(0.0))	0(0.0))	182(72.8)	68(27.2)
People with asthma cannot do as much physical exercise as other people.	0(0.0))	0(0.0))	0(0.0))	182(72.8)	68(27.2)
Asthma can be cured.	139(55.6)	111(41.9)	0(0.0))	0(0.0))	0(0.0))
Asthma cannot be controlled.	0(0.0))	0(0.0))		68(27.2)	182(72.8)

**Table 9 healthcare-11-01886-t009:** Comparison of the attitudes score before, after 2 weeks and 12 weeks of health education.

Variable	Mean	95% CI		Χ^2^[df]	F[df]	*p* Value
		Lower	Upper			
**Pre score**	11.3 ± 1.56	11.10	11.49	250.0 (1, 3)	14,471.7 (2, 248)	**<0.001**
**After 2 weeks**	22.6 ± 0.51	22.49	22.62			
**After 12 weeks**	22.4 ± 0.84	22.73	22.96			

Chi-square test and RMANOVA test were carried out.

**Table 10 healthcare-11-01886-t010:** Practice of the asthmatic patients regarding asthma before and after health education.

Variable		Pre	After 2 Weeks	After 12 Weeks
Do you usually visit a physician when developing symptoms?	Yes	67 (26.9)	229 (92.0)	204 (81.9)
	No	182 (73.1)	20 (8.0)	45 (18.1)
Do you use nasal spray?	Yes	69 (27.7)	203 (81.5)	203 (81.5)
	No	180 (72.3)	46 (18.5)	46 (18.5)
Do you buy over-the-counter drugs without consulting a physician?	Yes	249 (99.6)	0 (0.00)	90 (36.1)
	No	1 (0.4)	250 (100)	160 (63.9)
Do you avoid house dust and smoke?	Yes	109 (43.6)	220 (88.0)	219 (87.6)
	No	141 (56.4)	30 (12.0)	31 (12.4)
Do you strictly follow the doctors’ instruction?	Yes	78 (31.2)	202 (80.8)	190 (76.0)
	No	172 (68.8)	48 (19.2)	60 (24.0)
Have you done physical work or exercise in the last two weeks?	Yes	110 (44.0)	141 (56.4)	156 (62.4)
	No	140 (56.0)	109 (43.6)	94 (37.6)
Many people tend to forget taking medication. Have you forgotten in the last two weeks?	Yes	158 (63.2)	31 (12.4)	93 (37.2)
	No	92 (36.8)	219 (87.6)	157 (62.8)
If the asthma symptoms get worse, I change my medication.	Yes	125 (50.0)	63 (25.2)	48 (19.2)
	No	125 (50.0)	187 (74.8)	202 (80.8)
Do you use a fan to remove smoke and steam during cooking?	Yes	107 (42.8)	235 (94.0)	220 (88.0)
	No	143 (57.3)	15 (6)	30 (12.0)
Do you use deodorants or perfumes?	Yes	250 (100.0)	141 (56.4)	220 (88.0)
	No	0 (00)	109,943.60	30 (12.0)

## Data Availability

All data were available upon request to the corresponding authors.

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
