# Peer review of "Assessing the Impact of Health Education Intervention on Asthma Knowledge, Attitudes, and Practices: A Cross-Sectional Study in Erbil, Iraq"

_healthcare, 2023, doi:10.3390/healthcare11131886_

Round 1
Reviewer 1 Report
The manuscript focused on the changes of awareness on the asthma knowledge attitudes and practices after health education based on a cross-sectional study. It could provide some information to realize the effect on the awareness improvement by health education. However, there are major problems.
1. The weight and content of research was not enough to support an academic paper, and the manuscript was lack of depth.
2. This study focused on the asthma patients, but I think this work would be much more meaningful if the healthy population who are sensitive to environmental risk factors, such as children and elder, were enrolled in the study.
3. The abstract was not meaningful enough. In the Introduction part, no research progress was provided, but which was very important for others to know this research area. In the Method part, it was too redundancy and lack of logic.
4. The essay writing was not standard. For example, in the abstract, what is the meaning of KAP, it should be complete spelling when it was first shown. There were lots mistakes in the spelling, such as line 168 in page 4, “924.8%” or “24.8%”?
5. Why the knowledge score after 12 weeks was lower than that after 2 weeks? Deep analysis and discussion should be conducted. Additionally, this result made me think about how about the score after 6 months or after 1 year of health education.
6. The result of Figure 1 to Figure 4 could be illustrated to single one table; Table 6 to Table 8 could be integrated into one single table.
The essay writing was not standard. For example, in the abstract, what is the meaning of KAP, it should be complete spelling when it was first shown. There were lots mistakes in the spelling, such as line 168 in page 4, “924.8%” or “24.8%”?
Author Response
The manuscript focused on the changes of awareness on asthma knowledge attitudes and practices after health education based on a cross-sectional study. It could provide some information to realize the effect on the awareness improvement by health education. However, there are major problems.
Point 1: The weight and content of research was not enough to support an academic paper, and the manuscript was lack of depth.
Response 1: Thank you for your feedback on the weight and content of our research. We understand the importance of providing sufficient evidence and depth in an academic paper. We carefully review our findings and methodology to ensure they are adequately supported, and we expand upon the depth of our analysis to strengthen the manuscript. We appreciate your insights and have worked diligently to address these concerns.
Point 2: This study focused on the asthma patients, but I think this work would be much more meaningful if the healthy population who are sensitive to environmental risk factors, such as children and elder, were enrolled in the study.
Response 2: Thank you for comments. Although children and elder were the primary target group but due to consent, sampling frame and ethical issue, we only focus on adult patients which include elder groups also.
Point 3: The abstract was not meaningful enough. In the Introduction part, no research progress was provided, but which was very important for others to know this research area. In the Method part, it was too redundancy and lack of logic.
Response 2: Thank you and we appreciate your valuable feedback and would like to address the concerns you raised in your comments. The abstract and Introduction sections have been revised to reflect the key findings and significance of the study. The Method section has been revised to provide a comprehensive overview of the existing literature and advancements in the research area. The intention is to present a clear and concise methodology that can be easily understood and replicated.
Once again, we appreciate your valuable feedback and we are committed to improving the quality and coherence of our paper to meet the highest standards.
Point 4: The essay writing was not standard. For example, in the abstract, what is the meaning of KAP, it should be complete spelling when it was first shown. There were lots mistakes in the spelling, such as line 168 in page 4, “924.8%” or “24.8%”?
Response 4: Thank you for your concerns and suggestions. We corrected the spelling and all typographical mistakes and also checked with a native English speaker from Birmingham University.
Point 5: Why the knowledge score after 12 weeks was lower than that after 2 weeks? Deep analysis and discussion should be conducted. Additionally, this result made me think about how about the score after 6 months or after 1 year of health education.
Response 6: Thank you for your valuable comments. We appreciate your consideration of education level as a significant factor in our research results. It is worth noting that similar outcomes have been reported in previous studies conducted in other countries that focused on health education interventions. These studies also found that immediately after a two-week health education program, participants demonstrated effective knowledge retention, translating into improved behavioral practices. However, over time, individuals tend to disregard small or unfavorable preventive tasks. We have indeed addressed this phenomenon in the discussion section, particularly in the last paragraph. To address this issue, we have highlighted the importance of continued health education through mass media and the distribution of informational leaflets as effective strategies.
Point 6: The result of Figure 1 to Figure 4 could be illustrated to single one table; Table 6 to Table 8 could be integrated into one single table.
Response 6: Thank you for your valuable comments and suggestions. While we acknowledge the benefits of consolidating figures into a single table, it is important to note that our manuscript already contains nine tables. Considering the substantial number of tables present, we believe that utilizing graphic representations for these specific sections of the results will offer enhanced visualizations of the study outcomes pertaining to these particular factors. By employing such graphic presentations, we aim to improve the clarity and accessibility of the data for readers.

Reviewer 2 Report
GINA report and other international asthma guidelines recommend collaboration between asthmatics and their health care providers. Undertaking teaching asthma programme resulted in better asthma control. Additionally Authors reported that both asthma knowledge had a significant positive relationship with asthma self-management behaviors variability.
Author Response
Points: GINA report and other international asthma guidelines recommend collaboration between asthmatics and their health care providers. Undertaking teaching asthma programmed resulted in better asthma control. Additionally, Authors reported that both asthma knowledge had a significant positive relationship with asthma self-management behaviors variability.:
Response : Thank you for your valuable feedback and positive appreciation. We are greatly influenced by this kind of positive and inspiring support.

Reviewer 3 Report
Manuscript entitled „Assessing the Impact of Health Education Intervention on Asthma Knowledge, Attitudes, and Practices: A Cross-Sectional Study in Erbil, Iraq" is an original article. Authors have presented the results of a study on a positive impact of specific asthma education on asthma knowledge, attitudes and practices. It was the cross-sectional study which included a sample of 250 asthmatic patients (mean age 37.5 y; 129 males). The study was conducted between October 2018 and July 2019.
Authors have presented their original results which in my oppinion are correctly presented and are of a considerable scientific interest. Ten informative tables and six figures were used. The discussion is properly written containing enough criticism. Relevant references are cited. Patient education is essential to prevent unnecessary concerns about asthma and asthma medicines. Thus, I believe that the manuscript and the results of the study are worth to be published. The manuscript is written correctly and, in my opinion, it can be published in the present form except that I have a few really minor comments:
· How the patients (subjects) were selected? At random? According to the order of presentation and hospitalization?
· The abbreviation or acronym "KAP" should be explained at the first mention, both in the manuscript (line 84) and in the abstract (line 14).
· Why the name of the city (Erbil) was chosen as a keyword? (line 29)
Author Response
Point 1: How the patients (subjects) were selected? At random? According to the order of presentation and hospitalization?
Response 1: Patients were selected according to the order of presentations and hospitalization.
Point 2: The abbreviation or acronym "KAP" should be explained at the first mention, both in the manuscript (line 84) and in the abstract (line 14).
Response 2: Thank you for your suggestions. We added the full form of KAP in the abstract and introduction chapters at the beginning.
Point 3: Why the name of the city (Erbil) was chosen as a keyword? (line 29)
Response 2: There is no study done in Erbil about the KAP of asthma patients for prevention in the war-torn region. So when someone wants to do research on asthma or air pollution in Erbil, it will easily come out.
